# Deciphering the neural signature of human cardiovascular regulation

Jorge Manuel[1]*, Natalia Färber[1], Darius A Gerlach[2], Karsten Heusser[2], Jens Jordan[2,3], Jens Tank[2†], Florian Beissner[1†]

[1]Somatosensory and Autonomic Therapy Research, Institute for Neuroradiology, Hannover Medical School, Hanover, Germany; [2]Institute of Aerospace Medicine, German Aerospace Center (DLR), Cologne, Germany; [3]Chair of Aerospace Medicine, University of Cologne, Cologne, Germany

**Abstract** Cardiovascular regulation is integral to life. Animal studies have identified both neural and endocrine pathways, by which the central nervous system adjusts cardiac output and peripheral vascular resistance to changing physiological demands. The outflow of these pathways is coordinated by various central nervous regions based on afferent information from baroreceptors, chemoreceptors, nociceptors, and circulating hormones, and is modulated by physiologic and behavioural state. In humans, however, knowledge on central cardiovascular regulation below the cortical level is scarce. Here, we show using functional MRI (fMRI) that at least three hypothalamic subsystems are involved in cardiovascular regulation in humans. The rhythmic behaviour of these systems corresponds to high and low frequency oscillations typically seen in blood pressure and heart rate variability.

*For correspondence:
ManuelSanchez.Jorge@mh-hannover.de

†These authors contributed equally to this work

Competing interests: The authors declare that no competing interests exist.

## Introduction

Given the importance of cardiovascular regulation for our daily survival, it is not surprising that the body has several redundant and mutually interacting systems for this task. Important representatives include the classical baroreceptor (arterial and cardiopulmonary) and chemoreceptor reflexes, neuro-endocrine systems, like the vasopressin, sympatho-adrenal, renin-angiotensin-aldosterone, and the more recently discovered leptin-melanocortin system (*Sawchenko and Swanson, 1981*; *Dampney, 1994*; *Hilzendeger et al., 2012*; *Salman, 2016*).

Much of our present understanding of these systems stems from animals experiments, while mechanistic studies in humans remain scarce, mostly for lack of non-invasive methods to assess subcortical brain activity. Furthermore, most studies have evaluated only one system at a time. We must assume, however, that all neural and neuroendocrine cardiovascular control systems are carefully orchestrated by the central nervous system to achieve optimal regulation. The control centres responsible for such orchestration are presumed to be located in the brainstem and hypothalamus. They are well characterised for the 'textbook' baroreflex, but much less for the other systems.

To study cardiovascular regulation in human subjects, we devised an MR-compatible lower body negative pressure (LBNP) chamber that simulates orthostatic stress via footward blood volume displacement (*Figure 1c*; *Goswami et al., 2008*). The pressure of −30 mmHg used in our experiment has been shown to recruit both neural and endocrine mechanisms of cardiovascular regulation via activation of cardiopulmonary and arterial baroreceptors (*Loewy, 1981*; *Mark and Mancia, 1983*; *Kimmerly et al., 2005*; *Salman, 2016*). While assessment of endocrine pathways requires invasive methods, neural cardiovascular regulation by the autonomic nervous system can be measured non-invasively as it leads to characteristic rhythms of heart rate and blood pressure variability (HRV, BPV). To assess these rhythmic changes, we recorded blood pressure and heart rate traces during the LBNP-fMRI measurements and subjected them to spectral analysis isolating two

**eLife digest** Stand up too fast and you know what happens next. You will feel faint as the blood rushes away from your head. Gravity pulls the blood into your legs, and your blood pressure drops. To correct this imbalance, the brain sends nerve impulses telling the heart to beat faster and the outer blood vessels to tighten. This is the autonomic nervous system at work. It is how the brain adjusts cardiac output, and quietly controls other internal organs in the body. It involves two key regions of the brain, the hypothalamus and the brainstem, and stimulates smooth muscles and glands around the body.

The cardiovascular system also responds to the demands of exercise, with the heart supplying fresh blood laden with oxygen and the blood clearing out waste materials as it flows around the body. Perhaps surprisingly, blood pressure and heart rate fluctuate even at rest. The heart beats faster when breathing in and slower when breathing out. People's blood pressure, the force that keeps blood moving through arteries, also oscillates in so-called Mayer waves that last about 10 seconds.

Much of the current understanding of the inner workings of the cardiovascular system – and how it is regulated by the brain – stems from animal experiments. This is because few attempts have been made to simultaneously measure how a person's brain and cardiovascular system work with enough detail to see how brain waves and cardiac oscillations might interact.

To achieve this, Manuel et al. have now measured the brain activity, pulse and blood pressure of twenty-two healthy people while they were lying down in an MRI machine. This revealed that three distinct parts of the hypothalamus regulate cardiovascular output in humans. These 'subsystems' communicate with each other and with the lower brainstem, which sits beneath the hypothalamus. Manuel et al. also observed that the rhythmic activity of these subsystems runs in sync with oscillations typically seen in heart rate and blood pressure.

With this work, Manuel et al. have shown that it is feasible to measure different systems of cardiovascular control in humans. In time, with further experiments using this new approach, the understanding of chronic high blood pressure and heart failure may improve.

---

frequency bands: low frequency blood pressure variability (LF$_{BPV}$, ~0.1 Hz), reflecting the so-called Mayer waves of sympathetic origin (*Julien, 2006*); and high frequency heart rate variability (HF$_{HRV}$,~0.28 Hz) reflecting the primarily vagally mediated respiratory sinus arrhythmia (*Billman, 2013*). Note that both the LF and HF bands lie at the top or above the canonical frequency range of the BOLD signal (<0.1 Hz, Figure 1i); although several studies have recently provided evidence for BOLD oscillations beyond this frequency range (*Chen and Glover, 2015*; *Lewis et al., 2016*).

Subcortical control regions of cardiovascular function are mainly located in the hypothalamus, comprising the paraventricular nucleus (PVN), lateral hypothalamic area (LH), arcuate nucleus (Arc), dorsomedial nucleus (DMH), and median preoptic nucleus (*Supplementary file 1*), and in the lower brainstem, comprising the nucleus of the solitary tract (NTS), rostral and caudal ventrolateral medulla (RVLM/CVLM), nucleus ambiguus (Amb), and the caudal raphe nuclei, that is nucleus raphe obscurus (ROb) and nucleus raphe pallidus (RPa) (*Supplementary file 2*; *Coote, 2004*; *Dampney, 1994*; *Loewy, 1981*; *Saper et al., 2015*; *Benarroch, 1993*). Since these areas are notoriously hard to investigate in humans due to their small size, depth within the skull, and physiological noise from surrounding vessels and cerebrospinal fluid (*Brooks et al., 2013*; *Beissner, 2015*), we devised a high-resolution functional MRI approach and preprocessing pipeline specifically optimised for subcortical imaging. In particular, we balanced spatial and temporal resolution ($2{\times}2{\times}2$ mm$^3$, 1.23 s) to distinguish neighbouring nuclei, while critically sampling respiratory frequencies. Preprocessing involved maximising anatomical specificity by applying advanced distortion correction, symmetric diffeomorphic image registration to a study template, and omitting any spatial smoothing. In contrast to common practice, we did not regress physiological fluctuations from our data to avoid removing meaningful signal from the cardiovascular regions we were interested in *Iacovella and Hasson, 2011*. Instead, we opted for a spatial noise correction approach (*Beissner et al., 2014*) as part of our group-level statistical analysis.

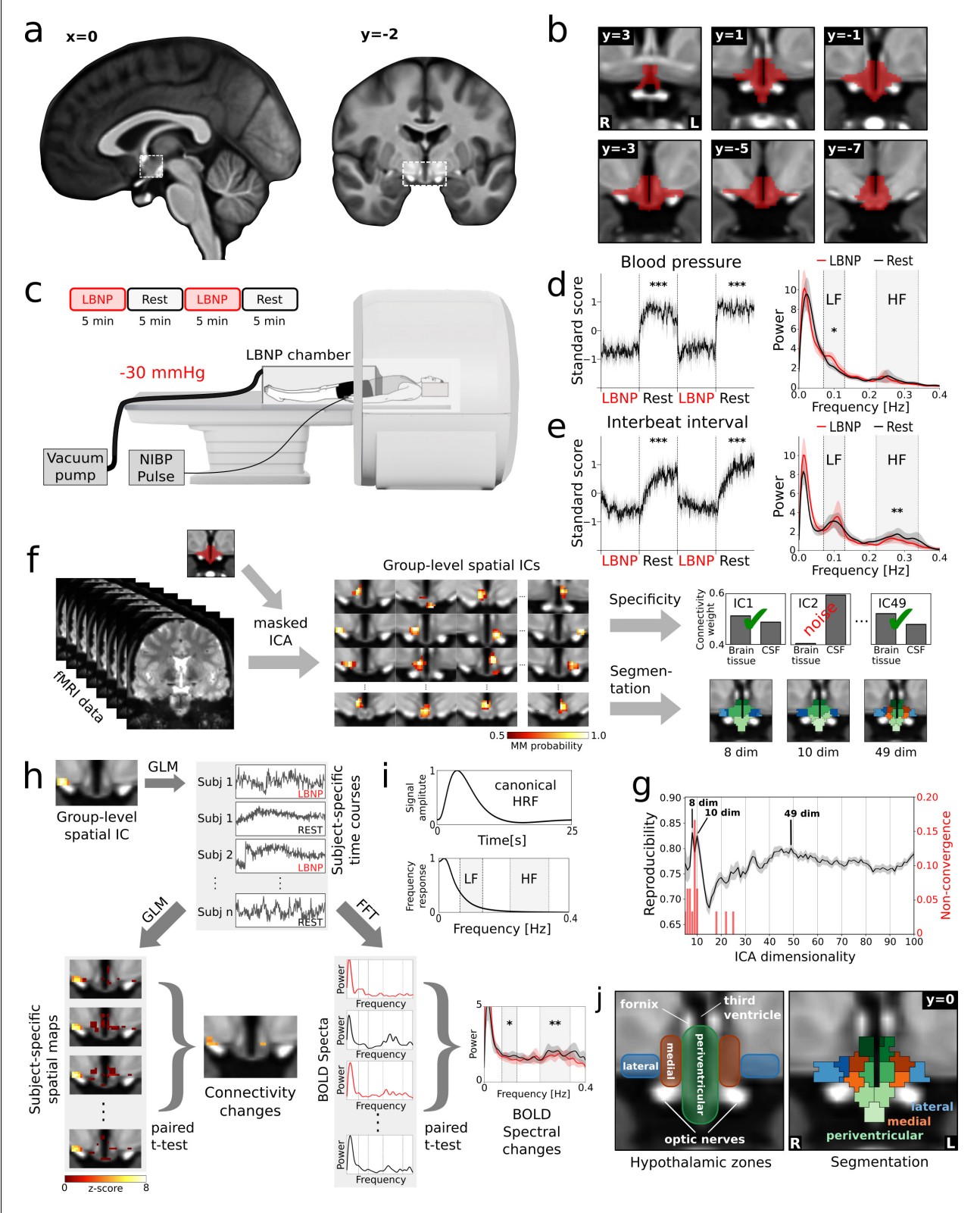

**Figure 1.** Identification of cardiovascular centres in the human hypothalamus. (**a**) Whole-brain T₁-weighted study template showing the anatomical localisation of the hypothalamus. (**b**) Anatomical mask of the hypothalamus. (**c**) Brain activity was recorded during alternating periods of lower body negative pressure (LBNP) and rest using functional magnetic resonance imaging (fMRI) with concurrent blood pressure and heart rate recordings. (**d,e**) Average beat-to-beat systolic blood pressure and interbeat interval decreased in response to LBNP ($t = -11.4$, p<0.001 and $t = -4.9$, p<0.001 resp.)

*Figure 1 continued on next page*

Figure 1 continued

with (d) blood pressure variability showing an increase of sympathetic low frequency (LF) oscillations (Mayer waves) ($t = 2.49$, $p<0.05$) and (e) heart rate variability showing a reduction of parasympathetic high frequency (HF) oscillations (respiratory sinus arrhythmia) ($t = 2.92$, $p<0.01$). (f) Spatially independent components (ICs), derived from masked ICA, were tested for specificity and used to segment the hypothalamus into functionally independent regions (see j). (g) The optimal number of ICs was 49 as evidenced by a bootstrap approach that maximised reproducibility ($r_{mean} = 0.80 \pm 0.01$), while penalising ICA non-convergence. (h) Spatial ICs were considered cardiovascular centres, when showing connectivity changes and spectral changes in the LF and/or HF range of the blood oxygenation level dependent (BOLD) signal during LBNP. Both measures were derived from subject-specific time courses either from a general linear model (GLM) or from a fast Fourier transform (FFT). (i) Both the LF and HF bands lie at the top or above the canonical frequency range of the BOLD signal as defined by the haemodynamic response function (HRF). (j) Final segmentation of the hypothalamus after removal of unspecific components showing three functionally distinct anatomical zones as expected from post-mortem anatomical studies (*Dudás, 2013*). Data in d-e and g-h are presented as mean with 95% confidence interval. All coordinates are in Montreal Neurological Institute (MNI) standard space. *$p{\leq}0.05$, **$p{\leq}0.01$, ***$p{\leq}0.001$.

The online version of this article includes the following figure supplement(s) for figure 1:

**Figure supplement 1.** Spatial independent components.
**Figure supplement 2.** Functional segmentation of the human hypothalamus.

## Results

LBNP led to a significant reduction of systolic blood pressure (Rest: $132.6 \pm 20.2$ mmHg, LBNP: $113.4 \pm 20.2$ mmHg, t(17)=-11.3, p<0.001, Cohen's d = $-0.95$) and a compensatory decrease of the interbeat interval (Rest: $0.95 \pm 0.12$ s, LBNP: $0.85 \pm 0.11$ s, t(20)=-4.9, p<0.001, d = $-0.96$), while measures of respiration (respiratory interval, respiratory volume, and respiratory volume per time) showed no significant changes. Furthermore, head movement was not significantly different between LBNP and Rest periods. We also observed increased spectral power of $LF_{BPV}$ (Rest: $0.14 \pm 0.03$, LBNP: $0.17 \pm 0.04$, t(17)=2.9, p=0.011, d = 0.97) and reduced spectral power of $HF_{HRV}$ (Rest: $0.18 \pm 0.08$, LBNP: $0.14 \pm 0.08$, t(20)=-2.4, p=0.0267, d = $-0.47$) indicating sympathetic excitation and vagal inhibition in response to the cardiovascular challenge, respectively (*Figure 1d,e*).

To identify cardiovascular control centres within the hypothalamus, we followed a two-step process; namely segmentation using masked independent component analysis (mICA) (*Beissner et al., 2014*) of the fMRI time series, followed by testing for cardiovascular involvement using two independent criteria. In the first step, the hypothalamus was segmented into 49 functionally distinct regions (*Figure 1—figure supplement 1*), followed by removal of six unspecific components (*Figure 1f*), leaving 43 components. The initial number 49 was derived by maximising mICA reproducibility, see *Figure 1g*. This high-dimensional segmentation yielded the characteristic three medio-lateral zones and three rostro-caudal regions expected from post-mortem anatomical studies (*Dudás, 2013*; *Figure 1j*, *Figure 1—figure supplement 2*, *Supplementary file 3*). The lower-dimensional decompositions with 8 and 10 subregions showed only two medio-lateral zones, and were thus not analysed. They were, however, consulted later to elucidate averaged intra-hypothalamic functional connectivity (see below). In the second step, each of the 43 (49 - 6) hypothalamic regions was tested for two criteria, namely LBNP-related changes in functional connectivity ($\Delta fc_{LBNP}$) with any region of the hypothalamus or lower brainstem, and changes in BOLD spectral power in one or both of the two cardiovascular frequency bands (*Figure 1h*).

Our analysis revealed five hypothalamic regions fulfilling both criteria for cardiovascular involvement (*Figure 2*, *Supplementary File 4*): the right anterior, and bilateral tuberal LH/supraoptic nucleus (SON, *Figure 2a–c*), the right tuberal PVN/posterior hypothalamic area (PH) (*Figure 2d*), and the arcuate nucleus (*Figure 2e*). Detailed information on the spatial distribution of BOLD spectral power in the two cardiovascular frequency bands is provided in *Figure 2—figure supplement 1*. All identified regions except the bilateral tuberal LH/SON showed positive $\Delta fc_{LBNP}$ with the lower brainstem (*Figure 2a,d–e*). In contrast, $\Delta fc_{LBNP}$ of the tuberal LH/SON was restricted to the hypothalamus. Here, both sides showed positive within-nucleus $\Delta fc_{LBNP}$, which can be interpreted as local activity changes, while only the left LH/SON showed additional negative $\Delta fc_{LBNP}$ with the arcuate nucleus (*Figure 2b–c*).

Further insights on hypothalamic functional connectivity of the identified regions came from the analysis of the low-dimensional ICA results (*Figure 3*). Here, inter-dimensional matching showed that right anterior LH/SON, PVN/PH, and Arc had low-dimensional equivalents (*Figure 3b,d*), while the

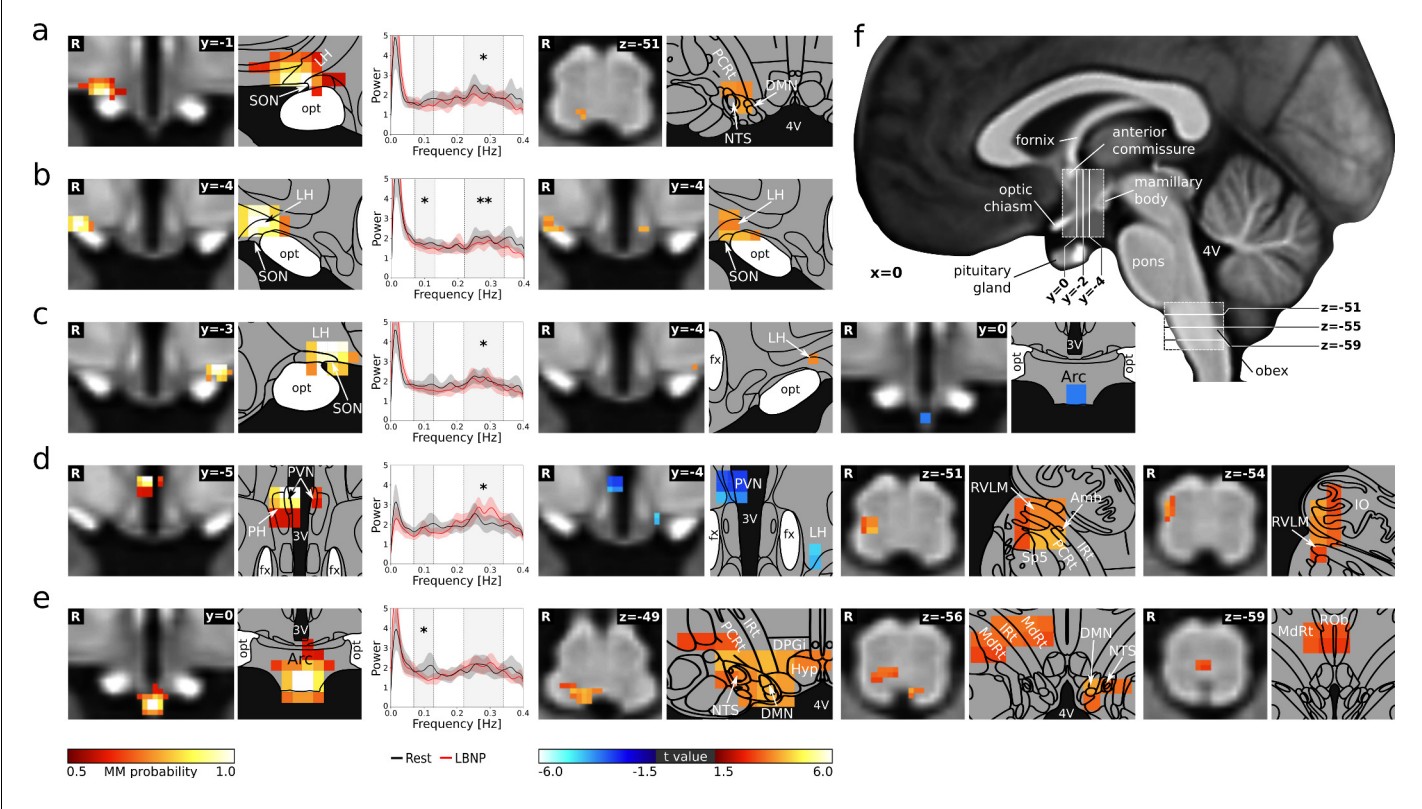

**Figure 2.** Neural signature of cardiovascular control. (a–e) Left: Individual hypothalamic centres derived from functional connectivity presented as mixture model thresholded probability maps. Centre: Normalised spectral BOLD signal changes (*n = 22*, paired t-test, p≤0.05). Right: Functional connectivity (fc) changes to the hypothalamus and lower brainstem (thresholded statistical maps, *n = 22*, paired t-test, p≤*0.05* family-wise error corrected). Warm/cold colours indicate an increase/decrease in fc during LBNP. A hypothalamic system of five regions showed both spectral BOLD and fc changes during LBNP. These cardiovascular centres included (a) the anterior part of the right lateral hypothalamic region (LH)/supraoptic nucleus (SON) showing reduced HF power (t = −2.2, p≤0.05) and increased fc with the dorsal lower brainstem including the nucleus of the solitary tract (NTS); (b–c) the bilateral tuberal part of the LH/SON showing reduced HF ($t_{right}$ = −2.93 $p_{right}$ ≤0.01, $t_{left}$ = −2.28 $p_{left}$ ≤0.05) and LF power ($t_{right}$ = −2.24 $p_{right}$ ≤0.05) and increased within-nucleus fc (both sides) as well as reduced fc with the arcuate nucleus (Arc, left side only); (d) the tuberal part of the paraventricular nucleus (PVN) showing increased HF power (t = 2.69, p≤0.05) and reduced within-nucleus fc as well as increased fc with the right lateral lower brainstem including nucleus ambiguus (Amb) and rostral ventrolateral medulla (RVLM); and (e) the Arc showing reduced LF power (t = −2.05, p≤0.05) and increased fc with the dorsal and midline lower brainstem including the NTS, dorsal motor nucleus of the vagal nerve (DMN) and nucleus raphe obscurus (ROb). (f) T₁-weighted study template showing slice localisation in the hypothalamus and brainstem. Spectral data in a-e are presented as mean with 95% confidence interval. All coordinates are in Montreal Neurological Institute (MNI) standard space. Atlas slices modified after *Mai et al., 2016* and *Paxinos et al., 2012* (with permission from the authors). *p≤0.05, **p≤0.01. Abbreviations: 3V, third ventricle; 4V, fourth ventricle; DPGi, Dorsal paragigantocellular nucleus; DMN, dorsal motor nucleus; fc, functional connectivity; fx, fornix; Hyp, hypoglossal nucleus; IO, inferior olivary nucleus; IRt, intermediate reticular nucleus; MdRt, medullary reticular nucleus; opt, optic tract; PCRt, parvicellular reticular nucleus; PH, posterior hypothalamic area; Sp5, spinal trigeminal nucleus.

The online version of this article includes the following figure supplement(s) for figure 2:

**Figure supplement 1.** Cardiovascular rhythms in the hypothalamus.

**Figure supplement 2.** Functional segmentation of the human hypothalamus without and with physiological noise regression.

**Figure supplement 3.** Functional connectivity changes during LBNP after physiological noise regression.

**Figure supplement 4.** Cortical connectivity of the cardiovascular hypothalamic regions.

**Figure supplement 5.** Comparison of the hypothalamic cardiovascular regions identified in this study and by *Li and Dampney, 1994*.

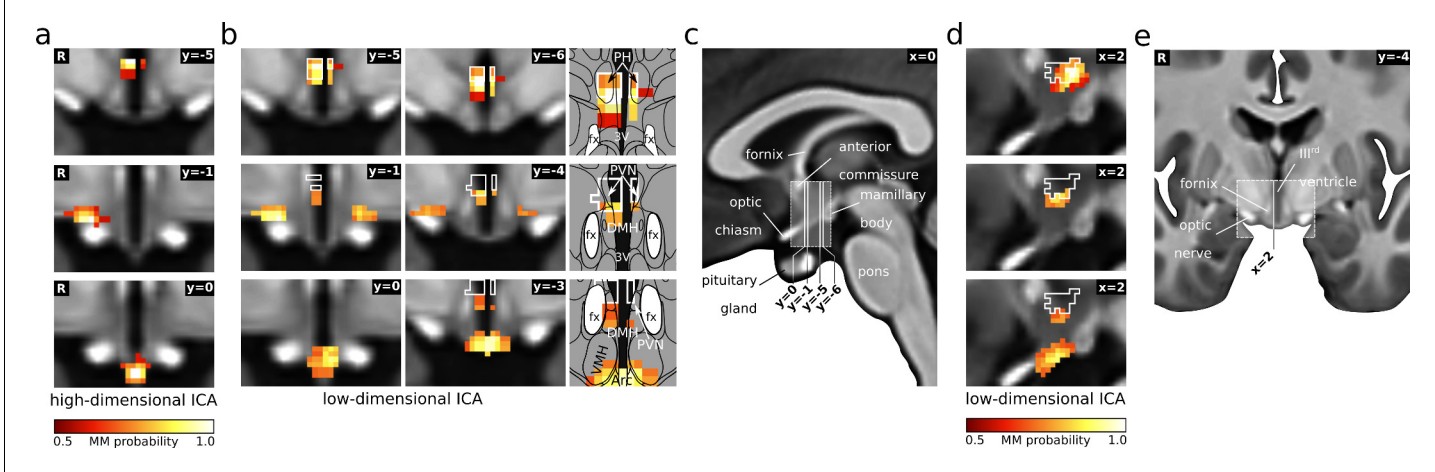

**Figure 3.** PVN/PH as a hub in cardiovascular regulation. (**a**) Three of the five hypothalamic cardiovascular centres derived from functional connectivity (*Figure 2*), namely PVN/PH (top), anterior LH/SON (middle), and arcuate nucleus (bottom) were also detected by low-dimensional independent component analysis (ICA) (**b**). Here, both LH/SON and arcuate components included portions of the PVN/PH and adjacent nuclei, such as the dorsomedial hypothalamic nucleus (DMH), indicating an important role of this region in cardiovascular regulation. For comparison, white lines in the coronal (**b**) and sagittal slices (**d**) indicate the outline of the high-dimensional PVN/PH component. In addition, the anterior LH/SON (middle) showed a bilateral symmetric shape in contrast to its strictly unilateral appearance in the original analysis. (**c,e**) $T_1$-weighted study template showing slice localisation in the hypothalamus. All independent components are presented as mixture model thresholded probability maps. All coordinates are in Montreal Neurological Institute (MNI) standard space. Atlas slices modified after *Mai et al., 2016* (with permission from the authors). Abbreviations: 3V, third ventricle; Arc, arcuate nucleus; DMH, dorsomedial hypothalamic nucleus; fx, fornix; PVN, paraventricular nucleus; PH, posterior hypothalamic area; VMH, ventromedial hypothalamic nucleus.

The online version of this article includes the following figure supplement(s) for figure 3:

**Figure supplement 1.** RVLM/Amb cluster increases functional connectivity to NTS during LBNP.

bilateral tuberal LH/SON did not. The low-dimensional versions of Arc and right anterior LH both included portions of the PVN/PH and adjacent nuclei.

Our supplementary analyses showed that our results are not driven by physiological noise. As the first analysis revealed, physiological noise regression had little effect on the masked ICA results as evidenced by the functional segmentation still showing the same known anatomical subdivisions of the hypothalamus (*Figure 2—figure supplement 2*). This was expected as we had previously shown that noise regression has little effect on masked ICA results (*Beissner et al., 2014*). Moreover, the ICs matching our five hypothalamic regions from the main analysis (spatial correlation r > 0.89) showed similar, yet smaller functional connectivity changes in response to cardiovascular challenge (*Figure 2*, *Figure 2—figure supplement 3*). The second supplementary analysis revealed that four of the five hypothalamic regions from the main analysis showed significant functional connectivity with cerebral regions that was clearly dominated by grey matter (*Figure 2—figure supplement 4*), while noise components would be expected to mainly show white matter or ventricular connectivity. The one remaining region was the PVN/PH, which showed very little cerebral connectivity at all.

## Discussion

We found five hypothalamic regions involved in cardiovascular regulation: the right anterior and bilateral tuberal LH/SON, the right tuberal PVN/PH and the arcuate nucleus. This selection of nuclei agrees with previous results from studies using Fos-like protein expression in response to prolonged hypotension (*Li and Dampney, 1994*; *Figure 2—figure supplement 5*).

Based on functional connectivity changes between the original hypothalamic regions and the lower brainstem (*Figure 2*), we were able to distinguish three major hypothalamic cardiovascular control systems (*Figure 4c–e*). The first was characterised by positive $\Delta fc_{LBNP}$ of the PVN/PH with a

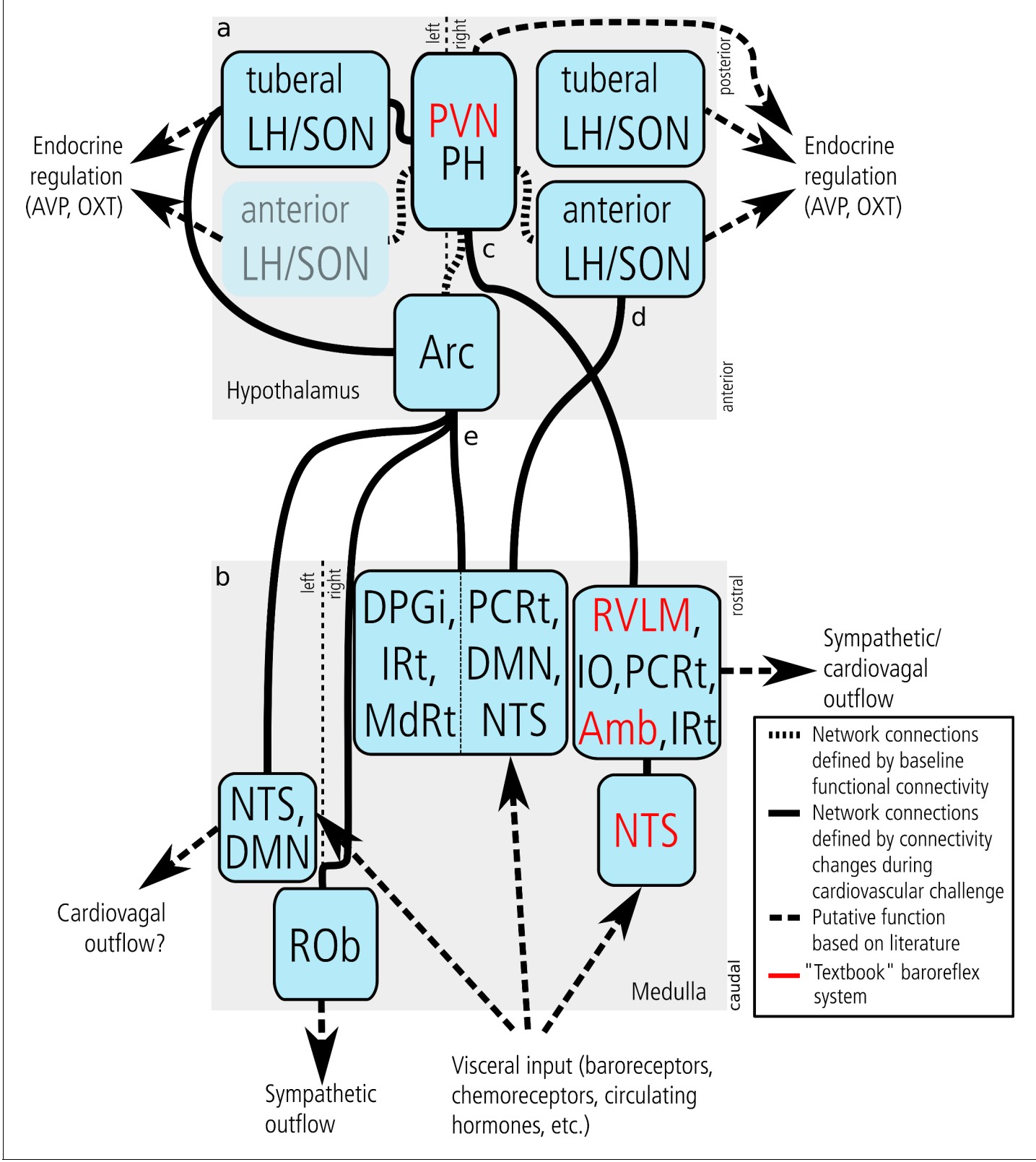

**Figure 4.** Central cardiovascular network. Summary of the network nodes and links in the hypothalamus (a) and lower medulla oblongata (up until the level of the vagal nerve root) (b) as defined by functional connectivity. The central cardiovascular network in humans comprises at least five hypothalamic and five medullary centres. Intra-hypothalamic connectivity emphasised the central role of the PVN/PH in cardiovascular regulation thus confirming results obtained in animals. Functional connectivity to the medulla oblongata was observed for three hypothalamic nuclei (c–e). Of these,

*Figure 4 continued on next page*

*Figure 4 continued*

the PVN/PH was connected to the 'textbook' baroreflex system (**c**) comprising the RVLM, Amb, and NTS (emphasised in red font). This system is known to be driven by baroreceptor input and regulate blood pressure by vascular and cardiac sympathetic, cardiovagal, and endocrine mechanisms (vasopressin and oxytocin release). In our experiment, it was strictly lateralised to the right medulla. (**d**) The second functional network connected the right anterior LH/SON with an ipsilateral medullary centre compatible with the NTS extending into DMN and PCRt. Projections of the LH to the NTS and DMN are well-established and mostly orexinergic, while likewise well-established projections from the NTS to the SON could trigger the release of vasoactive hormones from the pituitary gland. (**e**) The third and most widespread network emerged from the arcuate nucleus and comprised three distinct medullary regions: the left NTS and DMN, the midline ROb, and a large region in the right dorsal medulla comprising the NTS, DMN and several reticular nuclei. Although this system has received relatively little attention in comparison to the 'textbook' baroreflex system, there is clear evidence that the Arc plays an important role in integrating neural and hormonal signals from key cardiovascular organs and controlling sympathetic outflow. Abbreviations: Amb, nucleus ambiguus; Arc, arcuate nucleus; AVP, arginine vasopressin; DPGi, Dorsal paragigantocellular nucleus; DMN, dorsal motor nucleus of the vagal nerve; fx, fornix; Hyp, hypoglossal nucleus; IO, inferior olivary nucleus; IRt, intermediate reticular nucleus; LH, lateral hypothalamic area; MdRt, medullary reticular nucleus; NTS, nucleus of the solitary tract; OXT, oxytocin; PCRt, parvicellular reticular nucleus; PH, posterior hypothalamic area; PVN, paraventricular nucleus; ROb, nucleus raphe obscurus; RVLM, rostral ventrolateral medulla; SON, supraoptic nucleus; Sp5, spinal trigeminal nucleus.

region in the ipsilateral lateral medulla comprising the RVLM, inferior olivary (IO), parvicellular reticular nucleus (PCRt), Amb, and intermediate reticular (IRt) nuclei, as well as negative $\Delta fc_{LBNP}$ of the PVN/PH with itself (*Figure 2d*). We suggest that this system represents the 'textbook' baroreflex arc with its sympathetic (RVLM) and vagal arms (Amb); both under hypothalamic control by the PVN. Since baroreflex regulation classically involves the NTS and caudal ventrolateral medulla (CVLM), we conducted a supplementary mICA in the lower brainstem. This analysis yielded a functionally distinct region matching the above-mentioned RVLM cluster which was tested for $\Delta fc_{LBNP}$. As expected, a single cluster in the ipsilateral caudal NTS showed positive $\Delta fc_{LBNP}$ (*Figure 3—figure supplement 1*); however, we did not observe any functional connectivity changes with the CVLM. There is abundant evidence from animal experiments that the PVN projects heavily to the RVLM and Amb and to a lesser extent to Sp5 and reticular nuclei. These projections involve a wide variety of neurotransmitters including angiotensin-II, vasopressin, glutamate, and corticotropin-releasing hormone for RVLM, as well as oxytocin for the Amb (*Loewy, 1981*; *Coote, 2004*; *Geerling et al., 2010*; *Sapru, 2013*). Damage to the baroreflex arc elicits profound abnormalities in human blood pressure control (*Biaggioni et al., 1994*). Therapeutically, electrical baroreceptor stimulation has been tested for treating arterial hypertension and heart failure, although patients showed disparate treatment outcome (*Heusser et al., 2016*). Thus, a deeper knowledge of central baroreflex control may help understand inter-individual differences and identify patients most likely to respond to such therapies.

The second hypothalamic system (*Figure 2a +4d*) involved the right anterior aspect of the LH/SON showing positive $\Delta fc_{LBNP}$ with a small region in the ipsilateral dorsal medulla comprising the NTS, dorsal motor nucleus of the vagal nerve (DMN), and PCRt. These findings agree with animal experiments showing afferent connections of the LH/SON from the NTS (*Sawchenko and Swanson, 1982*; *Card et al., 2011*) and efferent projections to the NTS and DMN, the latter of which have been shown to be orexinergic (*Allen and Cechetto, 1992*; *Coote, 2004*). Their exact role in cardiovascular regulation, however, is still unclear. Potentially, this system also represents direct projections from the NTS to the SON that trigger the release of vasoactive hormones from the pituitary gland (*Grindstaff and Cunningham, 2001*). This pathway may complement neural cardiovascular regulation and serve as a backup for blood pressure maintenance in humans (*Jordan et al., 2000*). Clinically, hypotension-induced vasopressin release is attenuated in patients with neurodegenerative diseases affecting cardiovascular centres in the brainstem and hypothalamus (*Puritz et al., 1983*).

The third hypothalamic system involved the arcuate nucleus (Arc) (*Figure 2e +4e*), and showed positive $\Delta fc_{LBNP}$ with three different medullary regions. The first was located in the right dorsal medulla comprising NTS, DMN, PCRt, IRt and dorsal paragigantocellular nucleus (DPGi). The second included the right IRt, right medullary reticular nucleus (MdRt) and midline ROb. And finally, a third small region spanned the left NTS/DMN with its rostrocaudal position closely matching that of the contralateral NTS observed for the 'textbook' baroreflex control system (*Figure 3—figure supplement 1*). Interestingly, the role of the Arc in cardiovascular regulation has received relatively little attention despite several lines of evidence. The Arc has long been known to receive afferent fibres from the NTS (*Ricardo and Koh, 1978*). Changes in blood pressure and vascular resistance during

electric stimulation, Fos-like protein expression in response to prolonged hypotension and afferent renal nerve stimulation, and retrograde tracing studies link it to key cardiovascular organs (*Li and Dampney, 1994*; *Sapru, 2013*; *Rahmouni, 2016*). Furthermore, the Arc has been proposed as a potential region in which the leptin-melanocortin and renin-angiotensin-aldosterone systems interact to control sympathetic nerve activity (*Hilzendeger et al., 2012*), relevant to obesity-associated arterial hypertension (*Greenfield et al., 2009*). This issue is of utmost clinical relevance given the pandemic rise in the prevalence of obesity and associated cardiovascular disease.

This study combined functional connectivity and spectral analysis of fMRI signals to investigate hypothalamic regions involved during cardiovascular regulation. We found anatomically meaningful connectivity changes in five hypothalamic subregions during the cardiovascular challenge, all of which went along with spectral changes of the fMRI signal in the domains of low and high frequency cardiovascular regulation (LF,~0.1 Hz; HF,~0.28 Hz). In order to interpret these changes as BOLD signals, we need to assume that, at least for the HF domain, BOLD extends beyond the frequency range given by the canonical HRF (*Figure 1i*). Similar observations have recently been reported by several other groups (*Chen and Glover, 2015*; *Gohel and Biswal, 2015*; *Lewis et al., 2016*). However, caution is advisable, when interpreting results derived from BOLD signals. While the majority of our results were robust against the removal of physiological 'noise' signals, one should note that completely disentangling neuronal from non-neuronal BOLD signals is close to impossible for several reasons. Firstly, physiological noise is a mixture of several different noise sources, including those related to cardiac activity causing changes in arterial pulsatility, cerebral blood flow, cerebral blood volume and cerebrospinal fluid flow, and those related to respiration causing changes in the magnetic field and arterial carbon dioxide (*Brooks et al., 2013*). To complicate matters further, cardiac and respiratory activity are not independent but intricately linked through phenomena like the respiratory sinus arrhythmia. Secondly, several groups have recently shown that cerebral vascular regulation may be coordinated across long-distance brain regions, thus mimicking the structure of neuronal networks (*Bright et al., 2020*; *Chen et al., 2020*). Finally, every temporal regression of physiological 'noise' bears the risk of removing meaningful signal from the data (*Iacovella and Hasson, 2011*). Thus, the methodological approach used in this study cannot completely rule out the possibility that some of the connectivity we are seeing could be the result of highly structured physiological noise.

Nonetheless, we anticipate that the simultaneous in-vivo assessment of the different cardiovascular control systems in humans will deepen our understanding of these systems. Furthermore, access to individual neural signatures may facilitate development of novel pharmaceutical and electroceutical cardiovascular treatments, leading to an era of cardiovascular precision medicine.

## Materials and methods

The study took place at Hannover Medical School. It complied with the Declaration of Helsinki, and was approved by the local ethics committee (# 3404–2016). 22 healthy normotensive subjects ($24 \pm 5$ years, $22.1 \pm 3.1$ kg/m$^2$, $123 \pm 8/62 \pm 7$ mmHg, eight male) took part in the study after giving written informed consent and consent to publish their data anonimously.

We simulated orthostatic stress by means of lower body negative pressure (LBNP) (*Goswami et al., 2008*). Negative pressure of 30 mmHg was built up inside a custom-made polycarbonate chamber using a vacuum cleaner and a pressure gauge. The stimulation paradigm consisted of four alternating five-minute blocks, two with and two without negative pressure, starting with the negative pressure (*Figure 1c*). Functional MR images and physiological measures were continuously acquired during this cardiovascular challenge. Transitions between pressure states were excluded to avoid excessive movement. To quantify head motion, we calculated the root mean squares of relative image coordinate differences as transformed by the realignment matrices. We then checked for differences between fMRI runs with and without cardiovascular challenge by running a paired t-test.

### MRI data acquisition

All MR images were acquired on a Siemens 3T MAGNETOM Skyra using a 64-channel head/neck coil. The scanning protocol consisted of the following sequences (*Supplementary file 5*):

I.    Functional whole brain gradient-echo echo-planar images (EPI) ($T_R$ = 1230 ms; $T_E$ = 32 ms; 2 mm isotropic resolution; simultaneous multi-slice factor = 6; partial Fourier = 7/8; 4 × 233 volumes)

II.   Reference scan for motion correction and template formation; equivalent to (I) but without multi-band acceleration ($T_R$ = 7530 ms)

III.  Reference scans for unwarping: Two spin-echo images matched to (I) in distortion without multi-band acceleration; one with the same, the other one with inverted phase encoding direction.

IV.   $T_1$-weighted magnetisation-prepared rapid acquisition gradient-echo image (MPRAGE) ($T_R$ = 2300 ms; $T_E$ = 2.95 ms; $T_I$ = 900 ms; resolution: 1.1 × 1.1×1.2 mm$^3$, in-plane acceleration factor = 2)

## fMRI data preprocessing

fMRI data preprocessing was optimised for the brainstem and hypothalamus by avoiding superfluous resampling steps and unnecessary smoothing. On that account, motion correction (MCFLIRT [*Jenkinson et al., 2002*]) and unwarping (topup [*Andersson et al., 2003*]) were applied in a single transformation. Afterwards, brain extraction (BET [*Smith, 2002*]), grand mean scaling and high pass filtering (0.005 Hz) were applied. The data were not smoothed.

Two study templates were generated using Advanced Normalization Tools (ANTs [*Avants et al., 2008*]). The first one using the unwarped EPI reference images (II); the second one using the $T_1$-images. Functional images were transformed to the $T_1$-template for group analyses in a single transformation. This one-step procedure included a non-linear registration (symmetric diffeomorphism) to the EPI-template followed by a six-degrees-of-freedom transformation to the $T_1$-template. Finally, the results were transformed to Montreal Neurological Institute (MNI) standard space using a nonlinear transformation (ANTs). All transformations used linear interpolation during the resampling.

## Physiological measurements

The following physiological measures were acquired with an MR-compatible BIOPAC MP150 system:

I.    Respiration frequency and amplitude (respiration belt)

II.   Pulse (photoplethysmography)

III.  Non-invasive continuous blood pressure (pulse decomposition analysis of the digital artery pulse (CareTaker))

Pulse data were bandpass filtered (0.64–2.5 Hz) to remove scanner artefacts before running a semi-automated peak-detection. In the blood pressure recordings we first removed all dropouts and then applied a percentile filter (1$^{st}$-99$^{th}$ percentiles). Excluded data points were interpolated linearly.

## Data analysis

First, a masked independent component analysis (mICA [*Beissner et al., 2014*]) was performed on the concatenated functional data of all subjects (*Figure 1f*). We defined the hypothalamic mask for this analysis (*Figure 1b*) using the atlas of *Mai et al., 2016*. The ICA dimensionality of 49 was derived by test-retest reproducibility analysis in the range between 1 and 100 dimensions using 30 random split-half samplings (mICA toolbox [*Moher Alsady et al., 2016*]). After matching the components of both half samples by Hungarian sorting of their cross-correlation matrix, mean reproducibility was calculated. After excluding all values with ICA non-convergence, a dimensionality of 49 was found to maximise mean reproducibility ($r_{mean}$ = 0.80 ± 0.01) (*Figure 1g*). Independent components (ICs) were tested for specificity by running an unmasked dual regression (*Beckmann et al., 2009*) to a cuboid volume containing the brainstem and hypothalamus. We calculated the weighted quotient of activation in grey and white matter versus cerebrospinal fluid (probabilistic masks obtained using FAST [*Zhang et al., 2001*]). Components were considered unspecific if this quotient was smaller than one standard deviation from the mean. Using this measure, 6 components were excluded from further analysis. Assigning each voxel the component number with the highest z-value at that point yielded a segmentation of the hypothalamus into 43 functional centres (*Figure 1j*).

To discern which of the remaining 43 ICs were involved in cardiovascular regulation, we carried out two separate analyses: a spectral and a functional connectivity analysis. For the spectral analysis we defined two frequency bands, low frequency LF (0.1±0.03 Hz) and high frequency HF (0.28 ± 0.06

Hz) by looking at the spectral peaks of the autonomic recordings. The LF band was defined by the position of the Mayer peak in our subjects' blood pressure variability, whereas the HF band corresponds to the respiration frequency (mean ± standard deviation). We extracted the time series of every IC in every run and calculated their power spectral density for each frequency band normalised to the total power (similar to an fALFF analysis [*Zou et al., 2008*] at higher frequencies). Finally, we computed a non-parametric paired t-test to quantify spectral differences in the BOLD signal between LBNP and rest. In the functional connectivity analysis, we performed a dual regression to both, hypothalamus and lower brainstem. Ultimately, we calculated a non-parametric paired t-test thresholded at $p<0.05$ using family-wise error (FWE) correction with threshold-free cluster enhancement (TFCE) to identify functional connectivity changes between LBNP and rest. Components showing significant changes in both metrics, that is spectral and functional connectivity, were assumed to be involved in cardiovascular regulation. We overlaid the atlas of *Mai et al., 2016*, which is in MNI standard space, to our components and their connectivity changes in order to identify the involved nuclei. To avoid identifying only familiar nuclei, we report every major nucleus from the atlas overlapping with any voxel of a functional connectivity cluster.

Furthermore, to ascertain that our results were not driven by physiological noise, we conducted two supplementary analyses. Firstly, we repeated our initial analysis adding an additional physiological noise correction. Specifically, we applied slice-wise regression of cardiac and respiratory influences (first order, one interaction term) using FSL PNM (*Brooks et al., 2008*) before carrying out the masked ICA. We then matched the independent components to the ones of the original analysis by Hungarian sorting of the spatial cross-correlation matrix, and calculated their functional connectivity changes during LBNP. Since physiological noise regression influences frequency content in a non-linear manner, we did not test for spectral changes in this supplementary analysis.

Secondly, using dual regression and a non-parametric one sample t-test thresholded at $p<0.05$ (FWE corrected), we calculated functional connectivity of the identified hypothalamic regions with the whole brain (smoothed with a Gaussian kernel of 5 mm full width at half maximum). This was done to make sure that their functional connectivity was mainly in the grey matter and not in the ventricles or other regions, whose signals are driven by physiological noise.

## Acknowledgements

This work was supported by the Horst Görtz Foundation. The funding body had no role in study design, collection and analysis of data, and decision to publish.

## Additional information

### Funding

| Funder | Author |
| --- | --- |
| Horst Goertz Foundation | Florian Beissner |

The funding body had no role in study design, collection and analysis of data, and decision to publish

### Author contributions

Jorge Manuel, Conceptualization, Data curation, Software, Formal analysis, Visualization, Methodology, Writing - original draft, Writing - review and editing, Data acquisition; Natalia Färber, Data curation, Writing - review and editing, Data acquisition, Discussion of results; Darius A Gerlach, Jens Jordan, Methodology, Writing - review and editing, Discussion of results; Karsten Heusser, Conceptualization, Methodology, Writing - review and editing, Discussion of results; Jens Tank, Conceptualization, Supervision, Methodology, Writing - review and editing, Discussion of results; Florian Beissner, Conceptualization, Formal analysis, Supervision, Funding acquisition, Visualization, Methodology, Writing - original draft, Writing - review and editing

**Author ORCIDs**

Jorge Manuel (iD) https://orcid.org/0000-0003-1983-1448
Darius A Gerlach (iD) https://orcid.org/0000-0001-7044-6065
Karsten Heusser (iD) http://orcid.org/0000-0002-2571-5585
Florian Beissner (iD) http://orcid.org/0000-0003-0513-7551

**Ethics**

Human subjects: The study took place at Hannover Medical School. It complied with the Declaration of Helsinki, and was approved by the local ethics committee (# 3404-2016). All subjects gave written informed consent including consent to publish their data anonimously prior to participation.

**Decision letter and Author response**

Decision letter https://doi.org/10.7554/eLife.55316.sa1
Author response https://doi.org/10.7554/eLife.55316.sa2

## Additional files

### Supplementary files

• Supplementary file 1. Hypothalamic regions essential for cardiovascular control.

• Supplementary file 2. Medullary regions essential for cardiovascular control.

• Supplementary file 3. Coordinates and anatomical identification of spatial independent components.

• Supplementary file 4. Coordinates and anatomical identification of dual regression clusters of *Figure 2*.

• Supplementary file 5. Detailed MRI scan protocols. Note that the reference scans for unwarping (Reference SE (no SMS; AP and PA)) are the same sequence but with inverted phase encoding direction. Hence, they are summarised in one table here. However, both sequences need to be acquired for estimating the susceptibility induced field.

• Transparent reporting form

### Data availability

Raw data as well as necessary scripts for reproducing the results of this study are available at Zenodo.

The following dataset was generated:

| Author(s) | Year | Dataset title | Dataset URL | Database and Identifier |
|---|---|---|---|---|
| Manuel J, Färber N, Gerlach DA, Heusser K, Jordan J, Tank J, Beissner F | 2020 | fMRI during lower body negative pressure (LBNP) with concurrent physiological measurements | https://zenodo.org/record/3885042#.Xv2xGSi2mUk | Zenodo, 10.5281/zenodo.3885042 |

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
