## [Decision Letter]

**Acceptance summary:**

The reviewers and I are convinced that this study will help to progress our understanding of human cardiovascular control. We compliment the authors on the novel and innovative task and the carefully carried out analyses thereof.

**Decision letter after peer review:**

Thank you for submitting your article "Deciphering the neural signature of human cardiovascular regulation" for consideration by *eLife*. Your article has been reviewed by three peer reviewers, including Marc Tittgemeyer as the Reviewing Editor and Reviewer #1, and the evaluation has been overseen by Christian Büchel as the Senior Editor. The following individuals involved in review of your submission have agreed to reveal their identity: Ivan De Araujo (Reviewer #2); Olivia K Faull (Reviewer #3).

The reviewers have discussed the reviews with one another and the Reviewing Editor has drafted this decision to help you prepare a revised submission.

Summary:

All reviewers agree that the paper describes a very interesting study to explore brain centers of cardiovascular regulation in humans.

The regulatory pathways underlying neural responses to bodily signals are currently intensively researched particularly regarding the capacity of the organism to shape its ongoing activity. Relating to interoception (sensing of internal bodily states) brain-heart interaction came recently into focus to better understand the influence of internal sensory information on perception, memory, cognition, and gating of our fear responses. To that end, the topic of the paper is very timely, and the study may offer important insights into human neural pathways within cardiovascular control that may also lead to potentially beneficial clinical applications.

To carry out their study the authors have undertaken a clever experimental paradigm that makes use of functional MRI to characterize human subcortical brain regions involved in cardiovascular tone. The authors stimulated orthostatic stress via footward blood volume displacement (-30 mmHg), which induced sympathetic excitation and vagal inhibition in response to the challenge. The authors describe three hypothalamo-medullary subsystems potentially involved in cardiovascular control, preserving some parallel to activation patterns observed in animal studies.

A further interesting aspect of the paper is an elaborate processing pipeline for the notoriously difficult task to analyze brainstem and hypothalamus fMRI data.

Essential revisions:

While the study may thus advance our knowledge and all reviewers are in general enthusiastic about this work, some concerns must be adequately addressed before the paper can be accepted.

1) Relating to confounds associated with the involvement of control systems that only indirectly link to cardiovascular regulation per se: Specifically, the cardiovascular challenge employed induces responses in motor systems associated with respiration such as the craniofacial and thoracic motor systems. Several of those nuclei (parvocellular, intermediate, and medullary reticular nuclei, PCRt, IRt, MdRt) are known to be primarily involved in motor control of head and upper body muscles. It is unclear how exactly the authors did dissociate cardiovascular from other motor control systems in their analyses.

2) Various pathways implicate in the central regulation of cardiocvascular reactivity. The output of these pathways is coordinated by a number of central nervous system regions based on afferent information, but the central nervous system does not only involve the brain but also e.g. spinal cord. The authors mention for instance the melanocortine system which is not only involved in vagus nerve signaling but likely also regulated by somatosensory afferent from the spine via PBN. That said, it is sensible for the authors to emphasise primary regulators, such as hypothalamic regions and the lower brain stem. However, to then interpret the results as direct connections between hypothalamus and brainstem is a simplification, especially also as their connectivity measure is a very indirect one. This aspect needs to be discussed.

3) The authors argue that the "canonical" frequency of the BOLD signal lies outside the frequency bands of blood pressure and heart rate and that analysis is not affected by this. This argument needs more evidence.

4) Some consideration regarding susceptibility to noise in the image analyses is warranted. It is noticeable that many activated areas overlap with neighbouring ventricles, and in fact, some images suggest the peak activation voxel is centered at the ventricular area proper (e.g. Figures 2E, Figure 2—figure supplement 5D, E). Further assurance that no artefacts (e.g. movement given no spatial filtering etc.) influenced the determination of the areas described.

5) Furthermore, the use/non-use of physiological noise correction in this analysis needs more elaboration. While removing all signal associated with physiological recordings may drastically reduce the power to detect neural changes associated with the changes in physiology that are of interest, simply ignoring this step may dramatically influence the results. There would be two possible approaches that could ameliorate concerns that the results are driven by physiological artefacts (preferably both):

– A wider/whole-brain unmasked GLM analysis of the time-course associated with the 5 “cardiovascular” regions identified within the current analysis. The brain areas that correspond to these signals within a wider field of view would allow us to more accurately identify whether this signal is significantly associated with common physiological artefacts, as outlined here: https://doi.org/10.1016/j.jneumeth.2016.10.019

– A supplementary repeat of the analysis with the inclusion of physiological noise regressors, to understand which of the results can be robustly identified independently of the physiological artefacts.

6) The current method employed of only including ICA components that sit primarily within the grey matter does not fully prevent the influence of physiological artefacts, in particular second-order effects such as changes in relatively global signal primarily associated with cerebral vasculature, resulting from fluctuations in ventilation and/or metabolism (the former of which would occur with LBNP). The Discussion could also benefit from a short discussion of the limitations of disentangling physiological artefacts from neural underpinnings for the readers who are less familiar with the difficulties of this topic. This issue is particularly important if the authors are going to claim that their frequency analyses challenge the notion of the canonical HRF, as this claim can only be true if what they are reporting is in fact of neural origin, and not simply a non-neuronal artefact. Furthermore, the induction of LBNP is also regularly associated with a hyperventilatory response, and the high-frequency band sits right within the respiratory range and thus it is not surprising this sees differences with LBNP. Lastly, the HRF in the brainstem and hypothalamus are not clearly mapped nor understood, and as the vasculature in these smaller brain areas is vastly different from the original cortical locations where the HRF was identified, it might be advisable to proceed with caution when using strong statements in this regard.

7) The continuous blood pressure measurement seems to reside on peripheral arterial pressure pulse. Given that these are healthy participants, is there written consent and permission from the ERB explicitly to acquire arterial blood pressure, given that this is of course super invasive?

---

## [Author Response]

Essential revisions:While the study may thus advance our knowledge and all reviewers are in general enthusiastic about this work, some concerns must be adequately addressed before the paper can be accepted.1) Relating to confounds associated with the involvement of control systems that only indirectly link to cardiovascular regulation per se: Specifically, the cardiovascular challenge employed induces responses in motor systems associated with respiration such as the craniofacial and thoracic motor systems. Several of those nuclei (parvocellular, intermediate, and medullary reticular nuclei, PCRt, IRt, MdRt) are known to be primarily involved in motor control of head and upper body muscles. It is unclear how exactly the authors did dissociate cardiovascular from other motor control systems in their analyses.

We thank the reviewers for pointing out this potential weakness of our analysis. Further analyses of our physiological data have now revealed that neither respiratory interval, nor respiratory volume or respiratory volume per time were significantly affected by LBNP. On the other hand, we saw large (Cohen’s d≈1) and highly significant changes of cardiovascular parameters (blood pressure, BPV, heart rate, and HRV). Taken together, this indicates that LBNP’s effect on respiration were negligible in comparison to its cardiovascular effects. (In our experience, higher negative pressures are needed to induce respiratory changes like the ones described by the reviewers). Therefore, we believe that it is safe to assume that most of the connectivity we are seeing indeed reflects cardiovascular processes and not other motor functions, such as respiration. We have now added information on the absence of respiratory changes to the manuscript.

Of course, our findings do not completely rule out the possibility that respiratory or other motor nuclei were activated by LBNP. Indeed, the analysis of reflex processes always includes the possibility of stimulating further afferences or provoking further efferent activity in addition to those examined. This situation is very similar to co-activating autonomic areas in the cortex when using any form of stimulus (see Beissner et al., 2014). Although we are aware that some parts of the activations reflect autonomic reactions to the stimulus instead of higher cognitive functions, we usually cannot control for it.

Regarding our report of reticular nuclei, we would like to emphasize that we report every major nucleus from the atlas that overlaps with any voxel of a connectivity cluster. This is because we want to avoid making the mistake of only identifying familiar nuclei. Therefore, we have grouped the nuclei that belonged to the same cluster together in Figure 4. Our resolution clearly does not allow us to identify these individual nuclei in the first place. We have now added a clarifying sentence concerning this point to the Materials and methods section.

2) Various pathways implicate in the central regulation of cardiocvascular reactivity. The output of these pathways is coordinated by a number of central nervous system regions based on afferent information, but the central nervous system does not only involve the brain but also e.g. spinal cord. The authors mention for instance the melanocortine system which is not only involved in vagus nerve signaling but likely also regulated by somatosensory afferent from the spine via PBN. That said, it is sensible for the authors to emphasise primary regulators, such as hypothalamic regions and the lower brain stem. However, to then interpret the results as direct connections between hypothalamus and brainstem is a simplification, especially also as their connectivity measure is a very indirect one. This aspect needs to be discussed.

We thank the reviewers for this constructive criticism. We did not mean to imply that the connectivity was direct, although, as we discuss, there is ample evidence from animal studies for this assumption. We see, however, that the term “hypothalamo-medullary (sub-)system” may suggest a direct connection. To avoid confusion, we have now removed this term from the manuscript and exchanged it with “hypothalamic (sub-)system”. Furthermore, we made sure to only speak of functional connectivity throughout the manuscript.

3) The authors argue that the "canonical" frequency of the BOLD signal lies outside the frequency bands of blood pressure and heart rate and that analysis is not affected by this. This argument needs more evidence.

Unfortunately, the point tried to make here by the reviewers is not entirely clear to us. First of all, we do not speak of a frequency of the BOLD signal, but of a frequency range. This range is defined by the Fourier transform of the canonical hemodynamic response function (HRF) as shown in Figure 1i. Furthermore, we do not speak of the frequencies of heart rate and blood pressure (both around 1 Hz), but of heart rate variability (HF range, around 0.28 Hz) and blood pressure variability (LF range, around 0.1 Hz). However, we assume that these were just typos in the question.

Secondly, we do not say that frequency bands of HRV/BPV are generally outside frequency bands of the BOLD signal. This is only true for HF_HRV_ as shown in Figure 1i. For LF_BPV_, the same figure clearly shows overlap between the spectra (the light grey “LF” box overlaps with the non-zero part of the BOLD frequency response, while that of HF_HRV_ is considerably higher). See also: “Note that both the LF and HF bands lie at the top or above the canonical frequency range of the BOLD signal […]”.

Thirdly, in the sentence “that analysis is not affected by this” it is unclear, what the reviewers mean by “this”. We will try to give answers to both potential meanings. Should we have missed the main point, we ask the reviewers to clarify the critical points in a future re-review.

1) We do not claim that our analysis is not affected by cardiovascular (i.e. physiological) noise. We correct for this spatially by our masked ICA approach. However, since this was explicitly requested, we have now repeated our analysis using the more widely used temporal noise regression (see our reply to question # 5 below).

2) We do not claim that our analysis is not affected by BPV/HRV. On the contrary, what we are saying is that we find anatomically meaningful changes in functional connectivity of our hypothalamic subregions and these go along with spectral changes of the fMRI signal falling exactly into the BPV/HRV spectral region. In order to interpret these changes as BOLD signals, however, we need to assume that BOLD extends beyond the frequency range given by the canonical HRF. This assumption is not new, as the following papers show. They all find BOLD signals in higher frequencies than those defined by the canonical HRF. (Niazy et al., 2011; Chen and Glover, 2015; Gohel and Biswal, 2015; Lewis et al., 2016; Trapp, Vakamudi and Posse, 2018).

We have now added a detailed explanation about why we believe that what we are seeing are BOLD signals beyond the canonical frequency range to the beginning of our Discussion.

4) Some consideration regarding susceptibility to noise in the image analyses is warranted. It is noticeable that many activated areas overlap with neighbouring ventricles, and in fact, some images suggest the peak activation voxel is centered at the ventricular area proper (e.g. Figures 2E, Figure 2—figure supplement 5D, E). Further assurance that no artefacts (e.g. movement given no spatial filtering etc) influenced the determination of the areas described.5) Furthermore, the use/non-use of physiological noise correction in this analysis needs more elaboration. While removing all signal associated with physiological recordings may drastically reduce the power to detect neural changes associated with the changes in physiology that are of interest, simply ignoring this step may dramatically influence the results. There would be two possible approaches that could ameliorate concerns that the results are driven by physiological artefacts (preferably both):– A wider/whole-brain unmasked GLM analysis of the time-course associated with the 5 “cardiovascular” regions identified within the current analysis. The brain areas that correspond to these signals within a wider field of view would allow us to more accurately identify whether this signal is significantly associated with common physiological artefacts, as outlined here: https://doi.org/10.1016/j.jneumeth.2016.10.019– A supplementary repeat of the analysis with the inclusion of physiological noise regressors, to understand which of the results can be robustly identified independently of the physiological artefacts.

Since points (4) and (5) are closely related, we will answer them together. We very well understand the reservations the reviewers might have against our minimal preprocessing approach (i.e. standard realignment, no noise regression, no temporal low-pass filtering, no smoothing), when a large part of the field uses ever more aggressive methods of noise correction and data scrubbing.

Regarding the effect of motion, our original analysis only used fMRI data from the steady-state periods after the target LBNP pressure had been reached. This was done to avoid any contamination by movement artefacts during the pressure changes. To further rule out the possibility that head motion was driving our results, we have now quantified movement by calculating mean relative displacement for LBNP and Rest in all subjects. A paired t-test showed no significant differences (t = 0.05, p = 0.96). This result has now been added to the manuscript.

Regarding our decision to omit spatial smoothing, it is important to note that according to filter theory the smoothness of the image should match the smallest structure one aims to resolve. In our case, we are at the absolute resolution limit of our fMRI sequence, which is why spatial smoothing with any kernel would reduce our chances of identifying the nuclei of interest. Instead, we upsampled the data to twice the original resolution, which effectively increases smoothness without applying a smoothing kernel. Furthermore, we have previously shown that smoothing fMRI data in subcortical regions can impair data quality, since signals with high noise content (e.g., in the ventricles) are mixed with low-noise signals of adjacent regions of interest (Beissner et al., 2014). Finally, the non-parametric statistics we use do not rely on the central limit theorem and hence, do not need a certain smoothness of the data.

Regarding the use of physiological noise regression, we have previously shown that the results of masked ICA of noisy areas, like the brainstem, are not significantly altered by physiological noise regression (Beissner et al., 2014).

Nevertheless, we understand the need for more evidence that our results are not driven by noise. Therefore, we have now added results of two supplementary analyses as proposed by the reviewers.

Firstly, we repeated our initial analysis adding physiological noise correction. Specifically, we applied slice-wise regression of cardiac and respiratory influences using FSL PNM before carrying out the masked ICA. We then matched the independent components to the ones of the original analysis by Hungarian sorting of the spatial cross-correlation matrix and calculated their functional connectivity changes during LBNP.

As the analysis showed, physiological noise regression had little effect on the masked ICA results as evidenced by the functional segmentation still showing the same known anatomical subdivisions of the hypothalamus (new Figure 2—figure supplement 2). Furthermore, the ICs matching our five hypothalamic regions from the original analysis showed similar, yet smaller functional connectivity changes in response to cardiovascular challenge (new Figure 2—figure supplement 3).

Secondly, we calculated functional connectivity of the identified hypothalamic regions with the whole brain over all runs. This analysis revealed that four of the five hypothalamic regions from the original analysis showed significant functional connectivity with cerebral regions that was clearly dominated by grey matter (new Figure 2—figure supplement 4 ). In contrast, noise components would be expected to mainly show white matter or ventricular connectivity. The one remaining region was the PVN/PH, which showed very little cerebral connectivity at all (neither to grey matter nor to white matter or CSF).

We have now added the results of these additional analyses to the manuscript and are confident that our results mainly represent neuronal activity.

6) The current method employed of only including ICA components that sit primarily within the grey matter does not fully prevent the influence of physiological artefacts, in particular second-order effects such as changes in relatively global signal primarily associated with cerebral vasculature, resulting from fluctuations in ventilation and/or metabolism (the former of which would occur with LBNP). The Discussion could also benefit from a short discussion of the limitations of disentangling physiological artefacts from neural underpinnings for the readers who are less familiar with the difficulties of this topic. This issue is particularly important if the authors are going to claim that their frequency analyses challenge the notion of the canonical HRF, as this claim can only be true if what they are reporting is in fact of neural origin, and not simply a non-neuronal artefact. Furthermore, the induction of LBNP is also regularly associated with a hyperventilatory response, and the high-frequency band sits right within the respiratory range and thus it is not surprising this sees differences with LBNP. Lastly, the HRF in the brainstem and hypothalamus are not clearly mapped nor understood, and as the vasculature in these smaller brain areas is vastly different from the original cortical locations where the HRF was identified, it might be advisable to proceed with caution when using strong statements in this regard.

We agree that our approach does not completely rule out the possibility that some of the connectivity we are seeing could be the result of highly structured physiological noise. However, in this, our study is not different from any other fMRI study. The past few years have shown that we have to completely rethink the BOLD signal and its relationship with physiological “noise” as well as its exact role in functional connectivity. Nevertheless, we have now removed our statement about non-canonical BOLD signals from the Abstract and also toned down the statement in the caption of Figure 1. As requested, we have further added a short discussion of the limitations of disentangling physiological artefacts from neural underpinnings to the Discussion section.

Regarding a potential hyperventilatory response to LBNP, we have no evidence for a significant effect of 30 mmHg LBNP on respiratory interval, respiratory volume, and respiratory volume per time (see our response to point 1 above). This has been added to the manuscript.

Regarding the HRF in non-cortical regions, we agree that it is not well understood. However, we do not consider this a specific problem for non-cortical regions, as Rangaprakash et al., 2017, have shown that HRF variability can lead to strong pseudo-connectivity in cortical RSNs. Furthermore, Chen and Glover, 2015 have shown that the need for a narrower HRF with a higher frequency cut-off is evident in cortical fMRI when investigating the neuronal underpinnings of functional connectivity.

7) The continuous blood pressure measurement seems to reside on peripheral arterial pressure pulse. Given that these are healthy participants, is there written consent and permission from the ERB explicitly to acquire arterial blood pressure, given that this is of course super invasive?

This is a misunderstanding and we apologize for not being clear about this. Of course, the blood pressure measurement was carried out non-invasively by using an inflatable cuff on the thumb. We have now made it clear in the Materials and methods section, that our measurement was non-invasive.

**References**

Niazy RK, Xie J, Miller K, Beckmann CF, Smith SM. Spectral characteristics of resting state networks. *Prog Brain Res*. 2011;193:259-76.

Rangaprakash D, Wu GR, Marinazzo D, Hu X, Deshpande G. Hemodynamic response function (HRF) variability confounds resting-state fMRI functional connectivity. Magn Reson Med. 2018 Oct;80(4):1697-1713.

Trapp C, Vakamudi K, Posse S. On the detection of high frequency correlations in resting state fMRI. *Neuroimage*. 2018;164:202-213.